# GaLe: memory-efficient Global Approximate and Local Exact features

## Abstract

Embedded devices and Microcontroller units (MCUs) generally offer only a fraction of the memory and computational power available on machines equipped with general-purpose GPUs. Existing approaches for memory-efficient inference on these devices rely either on patch-based inference, which causes significant computational overhead, or approximation-based methods, leading to substantial accuracy degradation. In this work, we propose *GaLe*, a novel memory-efficient approximation technique that enables the deployment of pretrained deep neural networks on tiny, resource-constrained devices without the need for retraining. Our method introduces a feature map partitioning strategy that approximates layer outputs using two complementary representations: (i) a local exact ($L_E$) component that preserves fine-grained details and (ii) a global approximate ($G_A$) component that retains long-range dependencies. Differently from available tiling approaches, GaLe maintains compatibility with architectures with global receptive field operations and attention mechanisms, such as modern hybrid CNN-transformer models, while significantly reducing memory usage and computational overhead. We validate our approach on ImageNet classification, demonstrating performance comparable to exact inference methods while drastically reducing memory consumption and compute costs, achieving up to $65\%$ speedup on a `Cortex-M33` core for a $90\%$ RAM reduction compared to patch-based inference. Beyond efficient deployment, GaLe offers a general recipe for feature map decomposition, enabling the design of novel, resource-efficient convolutional and attention modules and potentially guiding memory-aware architecture search. We further demonstrate its versatility across classification, detection, and diffusion models, highlighting its potential as a foundation for future research on memory-efficient architectures. GaLe also benefits general-purpose GPUs, reducing the memory usage of diffusion models under 200MB (from 6GB) for high-resolution outputs. Code will be available upon acceptance.

## 1 Introduction

Deep neural networks have grown increasingly demanding in terms of memory and computational requirements, particularly with the rise of modern architectures such as attention-based models and transformers. These networks designs often rely on large intermediate activations and high parameter counts, pushing the limits of available hardware even in server- or GPU-class environments. The situation becomes even more challenging in the context of embedded and TinyML systems, where resources are severely constrained. Running models on such devices brings significant advantages in terms of privacy, latency, and energy efficiency, but it must contend with strict limitations:

- Static memory (FLASH): which sets the upper limit on model parameters ($< 2MB$);

- Dynamic memory (RAM): intermediate tensors must generally fit in less than $1MB$.

- Compute capacity: MCUs can tipically process tens of millions of operations per second, around 6 orders of magnitude less than GP-GPUs;

Among these, RAM capacity is particularly restrictive for modern building blocks such as inverted residuals (Howard, 2017; Sandler et al., 2018; Howard et al., 2019) and attention mechanisms (Cai

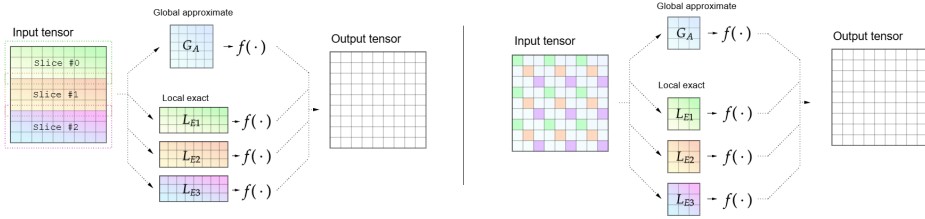

Figure 1: Graphical representation of our low-memory decomposition for convolutional networks (left) and hybrid networks (right) based on *local exact* and *global approximate* representations

et al., 2023; Liu et al., 2023; Dosovitskiy et al., 2021). To overcome this bottleneck, we propose a novel, memory-efficient approximation strategy that enables the deployment of deep networks on resource-constrained devices without requiring retraining. In fact, only a small calibration set is required to estimate the inference parameters. Our method can be applied directly to a pre-trained architecture and requires only the inclusion of resampling and striding operations during the forward pass, ensuring compatibility with most embedded platforms and runtimes (e.g., ONNX (Microsoft, 2025), TFlite (Google, b), nncase (Kendryte), STM32Cube.Ai (STMicroelectronics, a)). The proposed approach is based on *feature map partitioning*, where layer outputs are approximated by computing two complementary feature maps: (i) a *local exact* ($L_E$) representation for fine-grained details and (ii) a *global approximate* ($G_A$) representation to retain long-range dependencies. This approximation ensures compatibility with network architectures that are not typically supported by other memory reduction methods (Lin et al., 2021; Tan & Le, 2019; Patil et al., 2022; Hou et al., 2016), e.g. global receptive field operations (Hu et al., 2018) and attention blocks (Dosovitskiy et al., 2021), while ensuring smaller accuracy drops than comparable methods. Moreover, GaLe can be incorporated during training to guide architecture search or encourage networks to learn representations that are inherently memory-efficient. The main contributions of our work are two-fold: first, we analyze the limitations of existing memory reduction techniques, including mathematically exact methods and approximation-based approaches. Building on this, we introduce a novel hardware-aware adaptation of partial patch-based inference that optimizes memory usage through slices that are contiguous in RAM. We reduce the computational overhead of the patch-based approach by estimating the patch overlap required to reach a target precision, and incorporating global information through $G_A$ features to provide information where overlapping data is missing. Second, we extend our approach to transformer-based architectures, and in particular to modern hybrid models integrating both convolutional and attention-based mechanisms. We derive an attention formulation based on $G_A$ and $L_E$ feature maps that allows the operation to be performed sequentially on feature map blocks with less overhead compared to standard memory-efficient attention implementations (Rabe & Staats, 2021; Dao et al., 2022). Our formulation can either be used as a replacement, or work alongside these. The work is structured as follows: Section 2 introduces existing solutions for reducing resource usage in embedded scenarios, both from the architectural and inference-only standpoint; In Section 3 we describe our formulation for convolutional networks, analyzing different optimizations for the local exact slice-based inference (Section 3.1) and then introducing the global approximate representation (Section 3.4). We then analyze the application to attention mechanisms (Section 4) and hybrid architectures (Section 5). In Section 6, the achieved latency savings on four different embedded devices, the classification performance, and two case studies for application to object detection models and diffusion models are analyzed.

## 2 RELATED WORKS

**Training-based** approaches directly act on the network structure, generating optimized architectures with reduced computational complexity. For example, te MobileNet family of networks (Howard, 2017) and Xception (Chollet, 2017) were the first ones based on depth-separable approximation of standard 2D convolutions to reduce computation and number of parameters. MobileNetV2 (Sandler et al., 2018), MobileNetV3 (Howard et al., 2019) and MobileNetV4 (Qin et al., 2024) improved on this idea by using inverted residual blocks an automated network architecture search (NAS) to optimize the parameter/performance trade-off, and a hybrid transformer architecture. MCUNet (Lin

et al., 2020), MCUNet V2 (Lin et al., 2021) and V3 (Lin et al., 2022) specifically target small MCUs, while still using the inverted residual block architecture to optimize the number of parameters. EfficientNet (Tan & Le, 2019; 2021) introduced the concept of compound scaling, where the three main dimensions of a neural network (resolution, depth, and width) are jointly scaled to increase performance. PhiNets (Paissan et al., 2022) and XiNets (Ancilotto et al., 2023) introduced hardware-aware scaling, allowing disjoint optimization of memory, parameters, and latency in an MCU-targeted network. Recently, transformer-based efficient architectures have been developed, optimizing resources needed during training (Touvron et al., 2021) and inference (Liu et al., 2021; Mehta & Rastegari, 2023). Many ViT pruning strategies also rely on network fine-tuning to achieve good performance (Wei et al., 2023; Kim et al., 2022; Wang et al., 2024; Liu et al., 2024).

**Training-free** approaches can be used to reduce the memory of a pre-trained model. Reducing input resolution is the simplest among these approaches (Tan & Le, 2019; 2021). Since the memory consumption of intermediate feature maps scales quadratically with input resolution, this method effectively lowers RAM usage and computational cost. However, the performance degradation can be significant, particularly for tasks requiring fine-grained details. An alternative strategy is full patch-based inference (Akyon et al., 2022), where the input image is partitioned into smaller patches that are processed independently. This preserves the original resolution and keeps computational complexity similar to full-resolution inference. However, performance losses often arise due to the loss of global context across patches, making this approach task-dependent. In partial patch-based inference, introduced in TinyEngine (Lin et al., 2021) only the first layers of the network are executed patch-wise, before merging features and continuing inference at full resolution. This achieves mathematically identical results to full-resolution inference while reducing peak memory usage. However, the applicability of this tiling strategy depends on the architectural properties of the network - for example, it cannot be directly applied to networks with global receptive field operators such as attention (Zhao et al., 2020) or squeeze-and-excitation blocks (SE)(Hu et al., 2018). Many consumer inference engines for embedded devices rely on this kind of tiled execution for larger layers (STMicroelectronics, a; Lin et al., 2021; GreenWaves). Transformer training-free compression is usually achieved through token merging-based methods (Bolya et al., 2022; Bolya & Hoffman, 2023; Kim et al., 2024), where tokens are reduced at runtime to minimize the memory required by the attention map. The memory-efficient attention algorithm (Rabe & Staats, 2021) implemented by various consumer libraries (e.g. XFormers, Flashattention (Dao et al., 2022; Dao, 2023)) allows for a mathematically exact reduction of dynamic memory; however, it can cause significant computational overhead when large memory reduction ratios are required.

## 3 GaLe for Convolutional networks

Partial patch-based inference (PPBI) is widely adopted in consumer inference pipelines, as it can transparently reduce memory usage while ensuring exact numerical equivalence. However, it requires substantial computational overhead (as demonstrated in (Lin et al., 2021)) and is inherently limited to purely convolutional architectures that rely on small local receptive fields. Consequently, it is incompatible with architectures that use global receptive field operations, like SE modules (Hu et al., 2018), spatial pyramid pooling (He et al., 2015) or attention blocks (Mehta & Rastegari, 2023). To overcome these limitations, we propose a compute-efficient alternative that achieves memory reduction through a feature map slicing approach. Instead of relying on strict patching, we approximate the output of a convolutional block using two complementary representations (Figure 1):

- **Local Exact** ($L_E$): full-resolution features from a fraction of the original feature map
- **Global Approximate** ($G_A$): based on low-resolution features from the entire feature map

The proposed approach offers several advantages over traditional patch-based inference:

- **Reduced computation and latency:** The computational overhead drops from more than 100% to less than 20%, even for large memory reduction factors, while a hardware-aware tensor layout is used to decrease cache misses and increase throughput.
- **Broad applicability:** Our method extends beyond convolutional networks to architectures with global receptive field operations and attention mechanisms, including modern hybrid vision transformers, by introducing an optimized patching strategy tailored to these models.

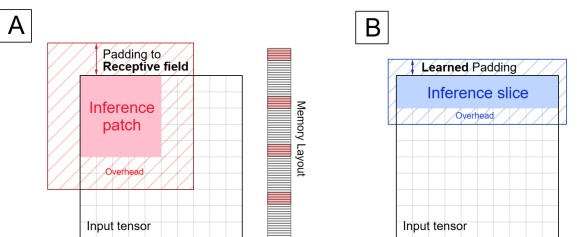

Figure 2: Proposed slicing vs classical patch-based inference. Our learned padding leads to significantly lower overhead compared to padding to the full receptive field. Moreover, for *NHWC* tensor inference, the proposed method relies on data stored in one continuous memory area, optimizing cache usage.

Although GaLe relies on mathematical approximations, it generally results in a minimal reduction in accuracy (typically less than $1\%$ for ImageNet classification). We argue that this is an acceptable trade-off in embedded environments; in fact, in such environments, accuracy is often more heavily impacted by other factors. Variability in vision sensors, the use of lower-quality optics, or simplified preprocessing pipelines — common in constrained environments - tend to dominate performance degradation, making minor model accuracy losses far less critical in the overall system.

### 3.1 COMPUTING LOCAL EXACT FEATURE MAPS

In partial patch-based inference, the input is divided into $N$ smaller, partially overlapping areas, with spatial dimensions $P_H \times P_W$ that are processed independently - reducing the size of intermediate activation tensors. To ensure accurate outputs at patch boundaries, patches must overlap sufficiently so that the *receptive field* of the layer at each output position is fully covered by valid input data. The required overlap $\mathcal{O}$ between patches is then equal to the receptive field of the deepest layer considered for patch-based execution: $\mathcal{O}_{ppbi} = \mathcal{R}_0(L)$.

Where $L$ is the layer index and $\mathcal{R}_0$ maps each layer to its receptive field on the input (layer 0). For example, considering a CNN composed only of convolutional layers with kernel size $k_i$, stride $s_i$ for layer $i$, receptive field size $\mathcal{R}_0(L)$ on the input layer (Araujo et al., 2019; Luo et al., 2016) can be computed as:

$$\mathcal{R}_0(L) = \sum_{l=1}^{L} \left( (k_l - 1) \prod_{i=1}^{l-1} s_i \right) + 1 \tag{1}$$

However, receptive field size from Equation 1 is only smaller than the input size for networks composed exclusively of local operations such as convolutional blocks and max-pooling layers. As such, PPBI can only reduce RAM requirements of a limited number of existing networks. In fact, this approach is not directly applicable to architectures that aggregate information across the entire input or operations with very large receptive fields, such as spatial pyramid pooling. Our approach generalizes partial patch-based inference to all network architectures by reducing the required overlap so that $\mathcal{O}_{GaLe} \leq \mathcal{O}_{ppbi}$, allowing for a user-controllable approximation error $\epsilon$ between the original and reconstructed feature maps. We rely on a *calibration pass* to determine the appropriate overlap $\mathcal{O}_i$ for each network layer patch. During calibration, for each layer $i$, we iteratively increase the overlap $\mathcal{O}_i$ by $\mathcal{R}_i(i+1)$ if layer $i+1$ is a convolutional layer, or $s_{i+1}$ otherwise. This process continues until either the MSE between the reconstructed output tensor and the original output tensor falls below $\epsilon$ or the overlap reaches the maximum allowed based on memory constraints. In this case, we increase the number $N$ of subdivisions for the current layer (algorithm in Appendix). Thanks to this iterative approximation strategy, the proposed method can approximate arbitrary network operators that may not be compatible with PPBI. When applied instead to architectures that can already be patched using traditional methods, GaLe can achieve significantly lower computational overhead, even for very small values of $\epsilon$.

### 3.2 MEMORY AWARE $L_E$ SLICING

The proposed method is based on dividing the input tensor into horizontal *slices* composed of multiple full rows ($P_W = W$) instead of square patches, as shown in Figure 2. From a hardware point of view, this leads to two important advantages:

Figure 3: (Left) Example of partial patch-based inference on 3 layers with 4 patches resulting in 12 kernel loading operations. (Right) Slice-based inference on 3 layers with 4 patches resulting in 6 kernel loading operations

- **Improved memory access patterns:** When working with tensors in NHWC format (the default for the majority of embedded runtimes (Google, b;a; STMicroelectronics, a)), slices align naturally with memory layout as shown in Figure 2. Each slice consists of values stored contiguously in memory, reducing cache misses and improving cache efficiency, resulting in lower latency, as shown in Section 6.

- **Processing during image acquisition:** Many vision sensors used in embedded devices (OmniVision Technologies, 2008) transfer image data row by row, scanning from top to bottom. This means inference can begin as soon as the first few lines are transferred, without waiting for the entire image to be transferred (or larger square patches), decreasing end-to-end latency and required memory.

### 3.3 OPTIMIZING MEMORY TRANSFERS

In PPBI, the number of patches is fixed and constant across all layers of the patch-based section, regardless of the memory requirements of individual layers. However, typical convolutional architectures require less memory the deeper in the network (often both tensor width and height are halved from one block to the next, while channel number doubles). This usually leads to unnecessary kernel loading operations, as deeper layers could fit the target memory even when split into fewer patches. Because of this, GaLe dynamically determines the number of patches for *each block* during the calibration pass, adapting to the memory footprint of the intermediate tensors. The adaptive slicing strategy allows deeper layers, which operate on smaller feature maps, to require fewer patches, thereby reducing redundant computations in overlapping regions. As a result, our method achieves lower computational overhead and faster inference compared to fixed-slice patching, especially in mobile-focused networks with aggressive downsampling in the early stages. Moreover, combining slices before the next layer operation allows reducing the total kernel loading operations (as shown in Figure 3), thus lowering the number of memory operations needed and, as such, network latency. Compared to standard PPBI, GaLe offers significant on-device latency savings (Figure 5), especially for simpler MCUs without a tiered memory structure.

### 3.4 $G_A$ FEATURE MAPS MAINTAIN GLOBAL INFORMATION

To overcome the information loss due to $L_E$ patch execution, we introduce a global representation through a Global Approximate feature map, which can be obtained simply from a scaled version of the input of a layer. Although resolution scaling performs worse than the alternatives in terms of the accuracy/RAM trade-off, this approach has two interesting aspects shown in Section 6:

- Requires significantly fewer operations than other alternatives for the same RAM.
- Preserves the global image information typically lost using other reduction methods.

For these reasons, we rely on this simple technique to obtain a Global Approximate ($G_A$) feature map representation. A low-memory approximation of the original tensor can be obtained by using a weighted combination of the local exact and global approximate factors, avoiding the loss of performance typical of other techniques. To construct the $G_A$ feature maps, we downsample by a factor $N$ (the same factor used to compute the $L_E$ feature maps) before passing the data through the most memory-intensive layers. We then upsample by the same factor to reconstruct an approximation of

| Feature Maps | Accuracy (%) |
|---|---|
| Ga + Le | 80.66 |
| Le only | 80.34 |
| Ga only | 79.16 |

Table 1: Comparison of feature map configurations on accuracy.

the original feature map. A weighting factor $\alpha$ is used when merging the local and global representations. While the computation of this additional feature map increases the processing overhead, this factor progressively decreases as $N$ increases, while, for lower reduction factors, $G_A$ computation can be omitted. As shown in Figure 5, compared to partial patch-based inference, we obtain significant reductions in latency, especially for high RAM reduction factors. Table 1 shows an ablation study on the effect of $G_A$ and $L_E$ feature maps on the performance of a MobileNetV4 network.

## 4 GaLe for Hybrid Networks

We extend the approach to modern attention-based and hybrid architectures, such as EfficientViT (Cai et al., 2023), FastViT (Vasu et al., 2023) and MobileNetV4 (Qin et al., 2024). As in the convolutional case, our method can offer superior performance than approximation-based techniques, with a fraction of the overhead required by math exact approaches based on memory-efficient attention (e.g. xFormers, FlashAttention), which recompute parts of the attention map multiple times during the forward pass. Our technique proves particularly effective for hybrid architectures, where both convolutional or attention layers can determine RAM peak usage. In fact, unlike existing approaches, GaLe can be used to reduce the memory of both the convolutional and attention-based parts in a unified framework. By analyzing RAM usage patterns in these models, we tailor GaLe to enable deployment on ultra-low-resource devices, such as MCUs, which are typically unsupported by existing methods. The proposed approach does not preclude the use of other memory-efficient attention implementations. Instead, it can be combined with them to balance approximation and computational overhead or to achieve higher RAM reduction factors, as demonstrated in Section A.1 with the application on diffusion models.

### 4.1 Attention through $L_E$ and $G_A$ approximation

To mitigate the large memory consumption associated with storing the attention map, while simultaneously avoiding the computational overhead introduced by recomputation-based methods, we propose an approximation scheme based on *block partitioning*. When using classical scaled dot-product attention on keys, queries and values $K, Q, V$ with feature dimension $d$, the operation is computed as:

$$\text{Attn}(Q, K, V) = softmax\left(\frac{\mathbf{A}}{\sqrt{d}}\right)V \quad \text{with } \mathbf{A} = QK^T \tag{2}$$

We approximate the attention map $\mathbf{A} \in \mathbb{R}^{N \times N}$ by dividing it into non-overlapping blocks and representing each block with a low-rank approximation:

$$\mathbf{A} = \begin{bmatrix} a_{11} & a_{12} & a_{13} & a_{14} \\ a_{21} & a_{22} & a_{23} & a_{24} \\ a_{31} & a_{32} & a_{33} & a_{34} \\ a_{41} & a_{42} & a_{43} & a_{44} \end{bmatrix} = \begin{bmatrix} \mathbf{A}_{11} & \mathbf{A}_{12} \\ \mathbf{A}_{21} & \mathbf{A}_{22} \end{bmatrix}$$

where each $\mathbf{A}_{ij}$ is a block representing the corresponding $b \times b$ subregion of $\mathbf{A}$, modeled as a weighted combination of the local exact component $\mathbf{L_E}$ and global approximate component $\mathbf{G_A}$:

$$\mathbf{A}_{ij} \approx \alpha\mathbf{L_E}_{ij} + (1-\alpha)\mathbf{G_A}_{ij} = \alpha v_{ij}\mathbf{Id}_b + (1-\alpha)C_{ij}\frac{\mathbf{J}_b}{b^2}$$

where $v_{ij} \in \mathbb{R}^b$ is a scalar controlling the local exact diagonal values, $C_{ij} \in \mathbb{R}$ is a scalar representing the global approximate constant value across the block, $\mathbf{Id}$ is the identity matrix and $\mathbf{J}$ the unit matrix, $\alpha$ the adjustable weighting factor. Using this approximation, it is possible to compute

an approximate form of $Attn(Q, K, V)$ by reducing the number of operations by a factor $b$ and the required RAM by a factor of $b \times b$ without computational overhead. We can compute $b$ local exact matrices $L_{E_k}$ containing the $k$-th values of $v_{ij}$, and one global approximate $G_A$ matrix with one value per block. Both $L_{E_k}$ and $G_A$ will have size $\frac{N}{b} \times \frac{N}{b}$. This structure allows us to compute the $softmax$ function in eq. 2 knowing only the values of one $L_{E_k}$ matrix, without the need to store the full attention map $\mathbf{A}$ in memory. For example, values along the 2nd row in $\mathbf{A}$ depend only on $L_{E_2}$ - without needing to keep the other $L_{E_k}$ in RAM. Thanks to this approximation, we can write a low-memory approximation of eq. 2 as

$$\tilde{\text{Attn}}(Q, K, V) = \alpha \cdot softmax\left(\frac{\sum_k L_{E_k}}{\sqrt{d_b}}\right)V + (1 - \alpha) \cdot softmax\left(\frac{G_A}{\sqrt{d_b}}\right)V \qquad (3)$$

In this way, we are approximating one attention operation on $N$ tokens as the weighted sum of $b + 1$ attention operations on $N/b$ tokens. Each of these can, in turn, be executed with memory-efficient attention algorithms for further memory savings.

## 4.2 Memory efficient inference

In inference, it is possible to compute eq. 3 efficiently by slicing $Q, K$ and $V$ into the b+1 components needed to calculate the local exact factors and the single component required for the global approximate factor, then performing the (b+1) attention operations between these. As mentioned, this leads to two advantages:

- **RAM savings.** The memory required for the proposed implementation is reduced by a factor $b^2$, as we are sequentially computing attention maps of size $N/b \times N/b$ instead of a single $N \times N$ matrix, requiring

$$\mathcal{O}\left(\frac{N^2}{b^2}\right) \qquad (4)$$

  It should be noted that, while this cost can become arbitrarily small depending on the factor $b$, RAM usage has a lower bound for the memory used to store the output of the attention operation (this is true even when using a memory-efficient attention implementation).

- **FLOPs savings.** The number of floating point operations required to perform the attention operation on $K, V, Q$ of size $N$ is $\mathcal{O}(N^2)$. In our case, this becomes

$$\mathcal{O}\left((b + 1)\left(\frac{N}{b}\right)^2\right) \approx \mathcal{O}\left(\frac{N^2}{b}\right) \qquad (5)$$

## 5 Hybrid Architectures

Recent state-of-the-art efficient neural network architectures often employ a hybrid design, combining convolutional layers in the initial stages with attention-based layers in the later stages (Qin et al., 2024; Vasu et al., 2023; Cai et al., 2023). In hybrid architectures, the primary contributors to peak RAM usage are typically two distinct components (Figure 4):

- **Initial Convolutional Layers:** Memory usage is usually the dominant factor in architectures that heavily downsample inputs early. This is especially true when high-resolution inputs are processed through multiple convolutional layers before reaching the attention stage. In this scenario, GaLe sliced inference is used to compute the early convolutional layers, to then progressively increase the number of split layers until the memory consumption falls below the target threshold, or until the attention layers become the dominant memory bottleneck.

- **High-Resolution Attention Layers:** When the attention layers operate on feature maps with large spatial dimensions, the memory peak can escalate rapidly. Modern hybrid architectures often reduce the spatial resolution progressively deeper into the network. For such models, we prioritize attention slicing on the layers that process the highest-resolution feature maps.

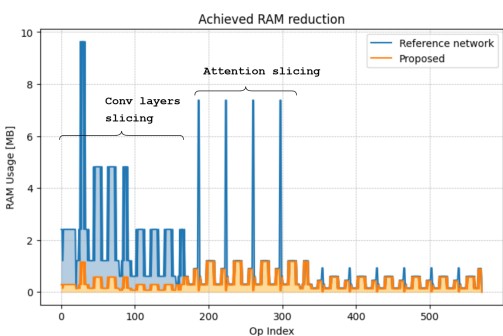

Figure 4: RAM requirements for EfficientViT B2 (Cai et al., 2023) hybrid architecture, and savings achieved through the proposed approach. The two factors contributing to peak RAM usage are clearly seen — the first convolutional layers acting on higher-resolution feature maps and the larger attention blocks.

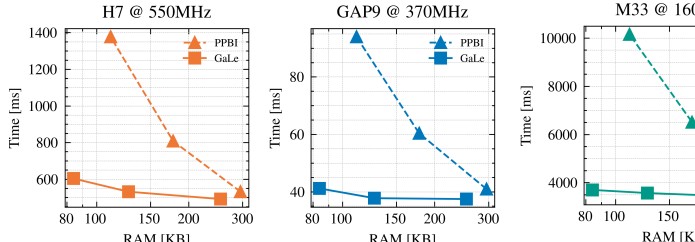
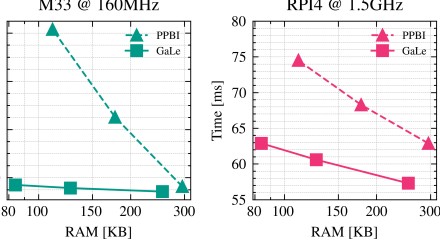

Figure 5: Latency vs RAM for `MobileNetV2` (256px, $0.5\times$, 1016 KB uncompressed memory). Compared with partial patch-based inference, our approach allows for significant RAM savings at lower latencies. This works across different hardware architectures, with stronger effects on devices with simpler memory architectures (65% latency reduction on M33 for a 90% RAM reduction, compared to 19% on a Raspberry Pi 4)

## 6 RESULTS

The proposed approach was benchmarked on four distinct hardware platforms: a high-performance Cortex-M7 MCU (`STM32H743`), a low-power Cortex-M33 MCU (`STM32U585`), a parallel RISC-V architecture (`GreenWaves GAP9`) and a Cortex-A72 MPU (`Raspberry Pi4`). We tested its performance on image classification across common networks used on embedded devices, both fully convolutional and attention-based. We compare our approach with resolution scaling, full patch-based inference (FPBI) and partial patch-based inference (PPBI) where applicable. We showcase GaLe in two case studies, object detection networks and image generation, in Section 6.1 and Section A.1, respectively. Additional results showing the behaviour of different GaLe configurations on various classification models are reported in Section C.1. Latency comparisons with PPBI are shown in Figure 5, while Table 2 shows the performance achieved by the different approaches. The proposed method consistently allows for lower RAM usage and better accuracy-to-overhead trade-offs. The three MobileNet variants highlight different behaviours with PPBI: MobileNetV2 ($\alpha = 1$) maintains the full network accuracy with an 88% RAM saving (albeit with a significant computing overhead), while our method can achieve an additional 40% RAM saving with lower overhead and only a tiny accuracy drop; MobileNetV3-Large does not allow for any memory reduction using PPBI due to the squeeze-and-excitation blocks in the first layers; MobileNetV4-Hybrid allows for some savings in the first layers, but memory peak becomes dominated by the attention layers. Similarly, token merging-based approaches fail to significantly reduce RAM peak for hybrid architectures, as this quickly becomes dominated by convolutional operations.

### 6.1 CASE STUDY: OBJECT DETECTION

We evaluated our approach in an embedded object detection task, assessing the deployment of two quantized object detection models with different resource footprints: RT-DETR-L (Zhao et al., 2023) (9.8MB RAM in INT8, $640 \times 640$) and YOLOv11n (Khanam & Hussain, 2024) (4.2MB RAM in INT8, $640 \times 640$). We analyze the deployment on two embedded platforms with limited computational resources: one with 2.5MB of RAM, representative of high-performance microcontrollers

| Network | | RAM [KB] | Top1% | Overhead% |
|---|---|---|---|---|
| | - | 6110 | 76.44 | - |
| | Res | 1527 | 64.16 | -74.7 |
| MobileNetV2 a110 | PBI | 1588 | 66.72 | +2.1 |
| | PPBI | 763 | 76.44 | +78.3 |
| | **GaLe** | **476** | 76.42 | +18.7 |
| | - | 4069 | 76.94 | - |
| | Res | 1017 | 62.50 | -73.9 |
| MobileNetV3 a100 | FPBI | 1017 | 66.04 | +7.3 |
| | PPBI | / | / | / |
| | **GaLe** | **317** | 76.56 | +16.3 |
| | - | 9010 | 80.46 | - |
| | Res | 1216 | 66.94 | -85.8 |
| ResNet50 | FPBI | 1261 | 71.96 | +5.1 |
| | PPBI | 1171 | 80.46 | +167.9 |
| | **GaLe** | **847** | 79.52 | +40.9 |

| Network | | RAM [KB] | Top1% | Overhead% |
|---|---|---|---|---|
| | - | 6314 | 81.80 | - |
| | Res | 1577 | 68.21 | -74.9 |
| MobileNetV4 HM | FPBI | 1623 | 69.08 | +1.6 |
| | ToMe | 6314 | 81.60 | -5.1 |
| | PPBI | 4922 | 81.81 | +26.4 |
| | **GaLe** | **884** | 80.66 | +14.6 |
| | - | 8342 | 81.96 | - |
| | Res | 1042 | 65.70 | -79.7 |
| FastViT | FPBI | 2085 | 74.60 | +5.4 |
| | ToMe | 8342 | 81.80 | -9.2 |
| | PPBI | 1668 | 81.96 | +43.4 |
| | **GaLe** | **992** | 81.10 | +5.9 |
| | - | 9732 | 82.46 | - |
| | Res | 1430 | 68.72 | -83.4 |
| EfficientViT | PPBI | 7687 | 82.46 | +31.4 |
| | ToMe | 9732 | 82.26 | -6.1 |
| | FPBI | 1625 | 72.21 | +4.1 |
| | **GaLe** | **1294** | 81.48 | +15.6 |

Table 2: Performance comparison of different models and methods on ImageNet classification. Accuracy and latency are reported, along with the relative computational overhead.

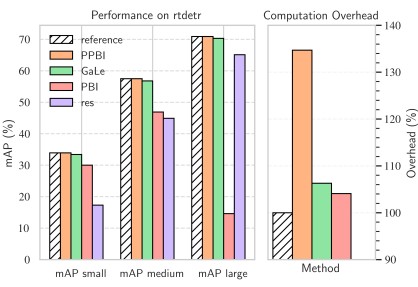 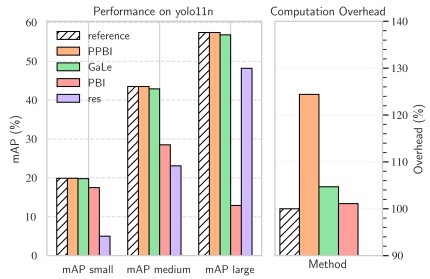

Figure 6: Comparison against alternative methods for reducing RT-DETR-L under $2.5MB$ of RAM (left) and YOLOv11n under $512KB$ of RAM (right)

(STMicroelectronics, b) and embedded accelerators (Technologies, 2021; Kendryte, 2018; Technologies, 2018), and another with 512KB of RAM, typical of low-power MCUs (STMicroelectronics, 2022). Meeting these constraints required a $74\%$ reduction for RT-DETR-L and $88\%$ for YOLOv11n. Figure 6 shows the performance achieved and the corresponding computational overhead. In both cases, lowering input resolution maintains performance for larger objects and reduces computational complexity, at the cost of significant performance losses on smaller objects. Full PBI instead preserves detection accuracy for small objects, but loses performance on larger objects due to spatial fragmentation. PPBI maintains the original detection accuracy across object sizes, although with substantial computational overhead. In contrast, GaLe achieves a negligible mAP reduction while maintaining minimal computational overhead, demonstrating an effective balance between efficiency and accuracy.

## 7 CONCLUSION

This work introduced GaLe, a novel approach for reducing memory consumption in convolutional and hybrid deep learning models, guaranteeing broad compatibility across various hardware platforms and network architectures. By leveraging a memory-aware feature map partitioning strategy that combines local exact and global approximate representations, GaLe enables the efficient deployment of pretrained models on memory-constrained embedded devices without retraining. In classification, our method achieved performance comparable to mathematically exact methods while significantly lowering memory usage. In object detection and image generation, GaLe outperformed existing approximation-based alternatives while requiring lower computational overhead than mathematically exact approaches. Additionally, when integrated with techniques such as token merging, GaLe enabled extreme memory compression for diffusion models while preserving image quality.

REPRODUCIBILITY STATEMENT

All novel results presented in this paper are fully reproducible using the code provided as supplementary material. Our implementation includes scripts for model evaluation, with default configurations matching those reported in the paper. Hyperparameter settings and evaluation protocols are detailed in the main text and appendix. For comparison results, we report numbers directly from the respective original papers.

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

# A APPENDIX

## A.1 CASE STUDY: DIFFUSION MODELS

We present an additional application of GaLe in the context of diffusion models as a replacement for self-attention layers to reduce memory consumption while avoiding the computational overhead associated with memory-efficient attention techniques (Rabe & Staats, 2021). To evaluate the effectiveness of our approach, we benchmark SD-Turbo (Sauer et al., 2024) comparing GaLe against memory-efficient attention and token merging (ToMe) for attention memory reduction. Additionally, we introduce a hybrid variant in which token merging is applied together with GaLe, computing $G_A$ through ToMe, while the $L_E$ features remain unchanged.

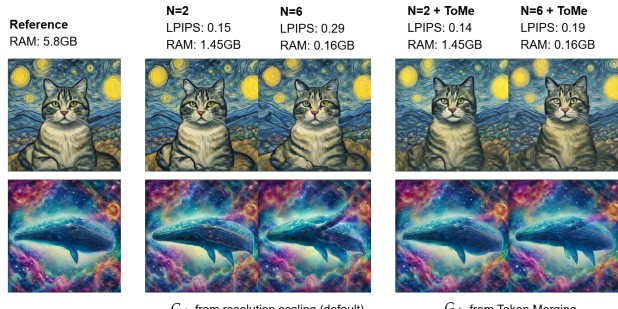

Figure 7: Our approach allows for significant RAM savings with minimal quality loss when applied to SD-turbo (Sauer et al., 2024). Token merging strategies can be used for computing the Ga feature maps, increasing performance.

As shown in Figure 8, GaLe achieves a substantially greater reduction in memory usage compared to ToMe alone, preserving the fidelity of generated images, as measured by Fréchet Inception Distance (FID) and Perceptual Similarity score (LPIPS (Johnson et al., 2016)) relative to the baseline network, while proving faster than standard memory-efficient attention. When GaLe is combined with ToMe, extreme memory compression factors can be achieved, which would otherwise be unattainable using ToMe alone, while maintaining a low FID score.

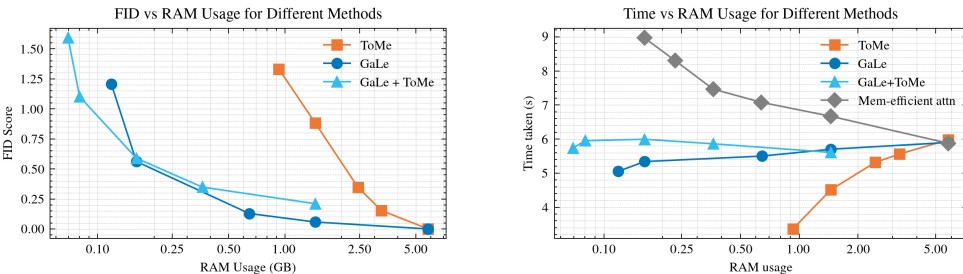

Figure 8: (Left) Time required for generating five $512 \times 512$ images on a mobile RTX3000 Ada. GaLe allows for significantly reduced RAM usage without the computational overhead of using memory-efficient attention. (Right) Compared to token merging approaches, GaLe allows for significantly reduced RAM usage. Token merging can be used for computing the Ga feature map, allowing for lower FID.

Figure 9 shows examples of images generated from SD-Turbo using GaLe, with the achieved RAM reductions and Perceptual Similarity scores.

Figure 9: Example images using sd-turbo with GaLe.

## A.2 DETAILED PLATFORM PERFORMANCE

Table 3 shows the real-world speedup achieved using GaLe compared to traditional partial-patch based inference. Higher speedups are obtained for simpler MCUs (eg `Cortex M33`), while more powerful devices with tiered caches (e.g. a `Raspberry Pi4`) do not benefit as much from the memory-aware slice layout.

| | N | RAM [KB] | H743 | GAP9 | M33 | RPi4 |
|---|---|---|---|---|---|---|
| Base | 1 | 1016 | 432 | 37.08 | 3132 | – |
| PPBI | 4 | 295 | 532 | 40.8 | 3632 | 62.9 |
| PPBI | 8 | 178 | 808 | 60.4 | 6512 | 68.3 |
| PPBI | 16 | 111 | 1378 | 93.96 | 10159 | 74.5 |
| GaLe | 4 | 254 | 492 | 37.48 | 3424 | 57.3 |
| GaLe | 8 | 127 | 532 | 37.8 | 3572 | 60.6 |
| GaLe | 16 | 84 | 604 | 41.2 | 3702 | 62.9 |

Table 3: Performance comparison of MBV2 A05 256x on different platforms. Times in ms

## A.3 GaLe calibration pass

The following pseudocode shows the calibration pass logic. Slice count is increased until the target performance are met. At each iteration, one forward pass is performed, the error is evaluated, and the overlap for the $i - th$ layer is increased.

---

**Algorithm 1** Calibration of slice Overlap Parameters

---

**Require:** Network, error tolerance $\epsilon$, total RAM
**Ensure:** Overlap values $\{O_i\}_{i=1}^{L}$
1: **for** $i = 1$ to $L$ **do**
2:     $O_i \leftarrow 0$
3:     $N \leftarrow ceil(\frac{RAM_i}{total\ RAM})$
4:     **repeat**
5:         **repeat**
6:             Run N-Le-based inference, overlap $O_i$
7:             Compute $\text{MSE}_i$ for output of layer $i$
8:             **if** Layer $i + 1$ is a convolutional layer **then**
9:                 $O_i \leftarrow O_i + \mathcal{R}_i(i+1)$
10:            **else**
11:                $O_i \leftarrow O_i + s_{i+1}$
12:            **end if**
13:        **until** $\text{MSE}_i < \epsilon$ **or** memory limit exceeded
14:        $N \leftarrow N + 1$
15:    **until not** memory limit exceeded
16: **end for**

---

# B  Sensitivity Analysis of Weighting Factor $\alpha$

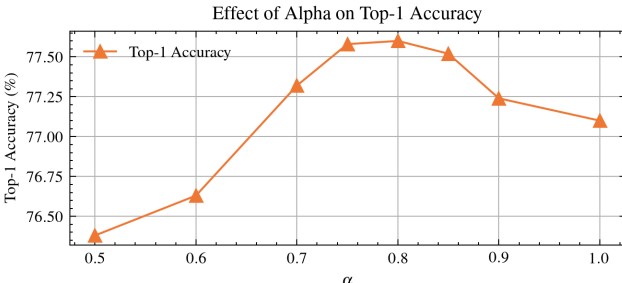

Figure 10: Effect of the weighting factor $\alpha$ on Top-1 Accuracy. The peak at $\alpha = 0.8$ demonstrates that a hybrid representation outperforms both pure local slicing ($\alpha = 1.0$) and aggressive global approximation ($\alpha = 0.5$).

To validate the necessity of our hybrid decomposition strategy, we conducted a sensitivity analysis on the weighting factor $\alpha$, which controls the fusion between the Local Exact ($L_E$) and Global Approximate ($G_A$) feature maps. As defined in the method, the approximated output is given by a weighted combination where $\alpha$ governs the contribution of fine-grained local details versus global semantic context. Figure 10 illustrates the impact of $\alpha$ on Top-1 Accuracy for ImageNet classification. Results indicate that the method is highly stable within the range $\alpha \in [0.7, 0.9]$, with peak performance observed at $\alpha \approx 0.8$. Notably, performance degrades at the extremes:

- At lower values (e.g., $\alpha = 0.5$), the model relies too heavily on the downsampled $G_A$ map, resulting in a loss of high-frequency details.

- At $\alpha = 1.0$ (pure slicing), the accuracy drops ($\approx 77.1\%$) compared to the hybrid peak ($\approx 77.6\%$). This confirms that when memory constraints prevent full receptive field overlap, the $L_E$ component alone suffers from insufficient global context, which is effectively recovered by the $G_A$ component.

## C    CALIBRATION EFFICIENCY AND SAMPLE SIZE

We analyzed the relationship between the size of the calibration dataset and the resulting model accuracy to assess data efficiency. As shown in Figure 11, the calibration process exhibits rapid convergence. The Top-1 Accuracy stabilizes significantly with as few as 16 to 32 samples, with negligible performance gains observed when increasing the sample size to 256. This data efficiency implies that the calibration step incurs minimal computational overhead and can be performed rapidly offline before deployment, without requiring large-scale validation sets or extensive processing time.

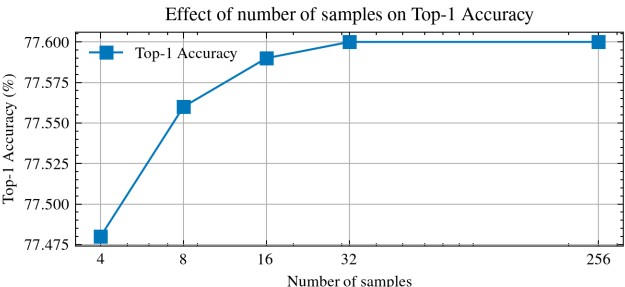

Figure 11: Effect of the number of calibration samples on Top-1 Accuracy. The method achieves optimal performance stability with as few as 32 samples, demonstrating high data efficiency and low setup overhead.

## C.1 ADDITIONAL RESULTS ON TIMM MODELS

Tables 4 and 5 show additional results achieved on common *timm* models using GaLe. Configurations are identified with the starting slice number ($S$), the number of split blocks ($B$), the allowed MSE error for the calibration pass ($\epsilon$) and the feature maps are used (for small RAM reductions, $L_E$ features already provide the same performance as the full network, so no $G_A$ features are used to reduce overhead).

| Network | Configuration | RAM Reduction % | Top-1 % | Overhead % |
|---|---|---|---|---|
| | - | - | 67.18 | - |
| | 2S-1B-$\epsilon$0.1-Le | 49.42% | 67.14 | +2.34% |
| MobilenetV2 a50 | 4S-2B-$\epsilon$0.1-Le | 73.62% | 67.08 | +5.86% |
| | 8S-3B-$\epsilon$1-Le | 85.86% | 67.42 | +9.90% |
| | 16S-4B-$\epsilon$5-GaLe | 93.75% | 64.88 | +10.94% |
| | - | - | 78.38 | - |
| | 2S-1B-$\epsilon$0.1-Le | 44.11% | 78.38 | +2.34% |
| MobilenetV2 a120 | 4S-2B-$\epsilon$0.1-Le | 72.16% | 78.38 | +7.42% |
| | 8S-3B-$\epsilon$1-GaLe | 85.16% | 78.38 | +19.53% |
| | 8S-3B-$\epsilon$1-GaLe | 85.07% | 78.38 | +19.53% |
| | 16S-4B-$\epsilon$5-GaLe | 92.33% | 78.24 | +34.38% |
| | - | - | 77.30 | - |
| | 2S-1B-$\epsilon$0.1-Le | 48.12% | 77.28 | +2.34% |
| MobilenetV2 a140 | 4S-2B-$\epsilon$0.1-Le | 73.15% | 77.28 | +5.86% |
| | 8S-3B-$\epsilon$1-GaLe | 85.26% | 77.32 | +14.32% |
| | 16S-4B-$\epsilon$5-GaLe | 92.16% | 77.28 | +27.34% |
| | - | - | 76.94 | - |
| MobilenetV3 Large a100 | 2S-1B-$\epsilon$0.1-Le | 48.44% | 76.94 | +2.34% |
| | 4S-2B-$\epsilon$0.1-Le | 72.98% | 76.94 | +5.86% |
| | 16S-4B-$\epsilon$5-GaLe | 92.13% | 76.56 | +16.30% |
| | - | - | 74.18 | - |
| | 2S-1B-$\epsilon$0.1-Le | 47.61% | 74.18 | +2.34% |
| MobilenetV3 L minimal a100 | 4S-2B-$\epsilon$0.1-Le | 74.18% | 74.18 | +5.86% |
| | 8S-3B-$\epsilon$1-GaLe | 85.05% | 74.16 | +15.36% |
| | 16S-4B-$\epsilon$5-GaLe | 91.36% | 74.18 | +32.23% |
| | 32S-5B-$\epsilon$10-GaLe | 95.31% | 72.56 | +39.06% |
| | - | - | 81.74 | - |
| | 2S-1B-$\epsilon$0.1-Le | 48.42% | 81.74 | +1.56% |
| MobilenetV4 Hybrid M | 4S-2B-$\epsilon$0.1-Le | 73.50% | 81.74 | +11.72% |
| | 8S-3B-$\epsilon$50-GaLe | 85.99% | 80.66 | +14.6% |
| | 16S-4B-$\epsilon$20-GaLe | 93.87% | 75.66 | +37.50% |
| | - | - | 80.86 | - |
| MobilenetV4 Hybrid L | 2S-1B-$\epsilon$0.1-Le | 48.48% | 80.86 | +1.56% |
| | 4S-2B-$\epsilon$0.1-GaLe | 73.48% | 78.61 | +18.72% |

Table 4: Additional results on timm models.

| Network | Configuration | RAM Reduction % | Top-1 % | Overhead % |
|---------|--------------|-----------------|---------|------------|
| ResNet18 | - | - | 72.56 | - |
| | 2S-1B-$\epsilon$0.1-Le | 42.25% | 72.54 | +14.06% |
| | 4S-2B-$\epsilon$0.1-GaLe | 60.80% | 72.50 | +32.03% |
| | 16S-4B-$\epsilon$5-GaLe | 90.72% | 64.12 | +13.28% |
| ResNet50 | - | - | 80.06 | - |
| | 2S-1B-$\epsilon$0.1-Le | 45.17% | 80.10 | +9.38% |
| | 4S-2B-$\epsilon$0.1-Le | 65.61% | 80.08 | +13.44% |
| | 16S-4B-$\epsilon$5-GaLe | 92.05% | 79.52 | +40.9% |
| ResNet200 | - | - | 83.60 | - |
| | 2S-1B-$\epsilon$0.1-Le | 43.82% | 83.60 | +12.50% |
| | 4S-2B-$\epsilon$0.1-GaLe | 62.39% | 83.60 | +40.62% |
| | 16S-4B-$\epsilon$20-GaLe | 90.66% | 81.94 | +45.12% |
| EfficientNetB0 | - | - | 78.50 | - |
| | 2S-1B-$\epsilon$0.1-Le | 48.54% | 78.54 | +2.34% |
| | 4S-2B-$\epsilon$0.1-GaLe | 70.05% | 78.54 | +2.81% |
| | 8S-3B-$\epsilon$1-GaLe | 85.98% | 78.54 | +14.58% |
| | 16S-4B-$\epsilon$5-GaLe | 92.34% | 78.30 | +21.88% |
| EfficientNetB2 | - | - | 80.18 | - |
| | 2S-1B-$\epsilon$0.1-Le | 47.59% | 80.18 | +3.91% |
| | 4S-2B-$\epsilon$0.1-Le | 62.50% | 80.14 | +18.75% |
| | 8S-3B-$\epsilon$1-GaLe | 84.35% | 80.12 | +19.53% |
| | 16S-4B-$\epsilon$5-GaLe | 92.07% | 79.46 | +23.83% |
| EfficientViT B1 | - | - | 80.06 | - |
| | 2S-1B-$\epsilon$0.1-Le | 48.46% | 80.10 | +2.34% |
| | 4S-2B-$\epsilon$0.1-Le | 72.76% | 80.08 | +5.86% |
| | 8S-3B-$\epsilon$1-Le | 85.85% | 80.00 | +12.50% |
| | 16S-4B-$\epsilon$5-GaLe | 92.86% | 78.56 | +12.70% |
| EfficientViT B2 | - | - | 82.90 | - |
| | 2S-1B-$\epsilon$0.1-Le | 48.47% | 82.88 | +2.34% |
| | 4S-2B-$\epsilon$0.1-Le | 72.62% | 82.92 | +7.42% |
| | 8S-3B-$\epsilon$1-GaLe | 85.86% | 82.90 | +15.10% |
| | 16S-4B-$\epsilon$5-GaLe | 93.07% | 82.72 | +20.12% |
| ResNext101 32x8 | - | - | 82.88 | - |
| | 2S-1B-$\epsilon$0.1-Le | 43.71% | 82.88 | +12.50% |
| | 4S-2B-$\epsilon$0.1-GaLe | 62.44% | 82.88 | +29.69% |
| | 16S-4B-$\epsilon$80-GaLe | 93.69% | 81.06 | +18.75% |

Table 5: Additional results on timm models.

