# OpenReview forum: "GaLe: memory-efficient Global Approximate and Local Exact features"
_ICLR.cc/2026/Conference — Submitted to ICLR 2026_

### Official Review · Reviewer_YTrW · 2025-10-20

**Soundness:** 3
**Presentation:** 3
**Contribution:** 2
**Rating:** 6
**Confidence:** 4

**Summary:**

This paper proposes the GaLe method to address the memory bottleneck in deep learning deployment on embedded devices. Its core innovation lies in the feature map partitioning strategy of "Local Exact (LE) + Global Approximate (GA)". The overall design is theoretically sound and practically applicable, yet there remain areas requiring supplementary verification or optimization.

**Strengths:**

1. Directly targets the memory constraint issue of embedded devices (MCUs) while avoiding the shortcomings of existing methods. It overcomes the high computational overhead of patch-based inference (PPBI) and mitigates the sharp accuracy drop of pure approximation methods, achieving a balance among "memory saving, accuracy preservation, and computational efficiency".
2. Supports direct deployment of pre-trained model and is compatible with CNNs, hybrid CNN-Transformer architectures, and operations with global receptive fields . It also adapts to mainstream embedded runtimes, lowering the threshold for industrial implementation.
3.Covers multiple hardware platforms  and multi-task scenarios with impressive key data—for instance, 90% memory saving and 65% inference acceleration on the Cortex-M33 core, and reducing the memory usage of diffusion models from 6GB to below 200MB, which is highly convincing.
4. Beyond deployment optimization, it provides a general solution for feature map decomposition, which can guide the design of resource-efficient convolution/attention modules and memory-aware architecture search, expanding the application scope of the method.

**Weaknesses:**

1. GA is generated based on "resolution scaling". Although this operation is lightweight, there is no comparison with more advanced global feature extraction methods, making it impossible to verify whether this strategy is the optimal solution for the "accuracy-efficiency" trade-off.
2. Under high memory compression ratios, the task-specificity of accuracy loss is not deeply analyzed—for example, in scenarios sensitive to details such as small object detection and high-resolution diffusion generation, whether the accuracy drop remains controllable.
3. The calibration phase adjusts the slice overlap (O) and slice number (N) iteratively to control errors, but the relationship between "calibration time" and "dataset size" is not explained. Given the limited computing power of embedded devices, excessive calibration time may undermine its engineering practicality.
4. While comparisons are made with PPBI, FlashAttention, and ToMe, the latest TinyML methods from 2024 to 2025 are not covered, making it impossible to clearly position GaLe in the current technical landscape.

**Questions:**

1. How is the weight α for fusing GA and LE determined? Is it a fixed value, adaptive per layer, or dynamically adjusted with tasks? Is there a systematic optimization strategy?
2.In CNN-Transformer hybrid architectures, how to automatically decide "which layers use LE slicing and which use GA approximation"? Does it rely on manual parameter tuning or an end-to-end layer selection mechanism?
3. When GaLe is combined with ToMe, GA is generated via ToMe while LE remains unchanged—does this combination cause consistency conflicts between local and global features? Are there optimization designs for the fusion logic?
4.After multiple rounds of inference on embedded devices, does the memory management of GaLe pose a leakage risk? Will accuracy drift occur under hardware interferences such as temperature and voltage fluctuations?
5.Can the calibration module and feature fusion module of GaLe be further lightweighted? For example, on MCUs, whether the memory usage and computational time of these two modules will become new bottlenecks?

**If the author can address my questions, I am willing to improve my rating.**

---

> ### Author Response · Authors · 2025-11-25
> **Response to Reviewer YTrW (1/2)**
>
> We are grateful for your comprehensive review. Below, we address your specific questions and concerns.
>
> **W1:** We selected resolution scaling (bilinear interpolation) not for its accuracy, but primarily because this operation is very fast to perform on embedded devices, reducing computational overhead. However, GaLe is designed as a general framework where the $G_A$ mechanism is interchangeable. As shown in our Diffusion Model case study (Appendix A.1), we successfully replaced the standard resolution scaling with Token Merging (ToMe) to generate the $G_A$ map. This swapped component allowed for extreme memory compression while maintaining image fidelity (FID/LPIPS), proving that developers can substitute more advanced, task-specific global extractors if their specific platform allows. While bilinear interpolation is the most generally compatible option for embedded devices, other task-specific solutions may exist depending on target hardware.
>
> **W2:** We agree that a detailed analysis of these edge cases is important. We performed two additional experiments to validate our approach in these conditions.
> **Small objects detection performance:** We explicitly analyzed this small object scenario in more detail here: [https://ibb.co/W4D9DzGV](https://ibb.co/W4D9DzGV)). Results show that GaLe maintains high accuracy across both small and large objects by combining the detail of $L_E$ with the context of $G_A$, even for extreme memory compression ratios.
> **Robustness at higher resolutions:** The following table demonstrates the applicability of the approach for high-resolution image generation at high compression ratios. We can see from the LPIPS score that image fidelity is maintained, and actually increases with higher resolution generation:
>
> | Attn split | LPIPS (512x512) | RAM (GB) | LPIPS (768x768) | RAM (GB) | LPIPS (1024x1024) | RAM (GB) |
> | :--- | :--- | :--- | :--- | :--- | :--- | :--- |
> | **1** | 0.00 | 5.77 | 0.00 | 12.98 | **0.00** | 23.07 |
> | **4** | 0.15 | 1.44 | 0.14 | 3.24 | **0.13** | 5.77 |
> | **9** | 0.16 | 0.64 | 0.20 | 1.44 | **0.09** | 2.56 |
> | **25** | 0.30 | 0.23 | 0.21 | 0.52 | **0.15** | 0.92 |
>
> **W3, Q5** We have analyzed the calibration efficiency to address this. As shown in the figure linked below and in the revised appendix, the calibration converges very rapidly. Top-1 Accuracy stabilizes with as few as 16 to 32 samples. ([https://ibb.co/N6V6211m](https://ibb.co/N6V6211m))
>
> Because calibration is data-efficient, it can be performed quickly even on embedded devices, with as few as 16 inference passes. Furthermore, this is typically an **offline compilation step** (performed on a host machine before deployment), meaning it does not consume cycles or memory on the MCU during runtime when executed before deployment. In this case, the already optimized model can be converted to ONNX or other formats, maintaining the optimized execution logic.
>
> **W4**  We acknowledge that tiled inference is a well-established concept for reducing memory in pure CNNs. However, GaLe was specifically developed to be orthogonal to these, while addressing the failure cases of these existing methods.
> * **Limitations of Prior Work:** Traditional block convolution methods fail when applied to modern architectures containing global operators (e.g. Squeeze-and-Excitation (SE) blocks or Self-Attention mechanisms) because these layers require information from the entire feature map, which is broken during standard tiling
> * **GaLe's Contribution:** Our novelty lies in the hybrid decomposition ( $L_E + G_A$ ). By introducing the Global Approximate ( $G_A$ ) component, GaLe restores the global context lost during tiling. This allows us to extend memory-efficient inference to architectures where it was previously impossible, such as MobileNetV3 (SE blocks) and Hybrid Vision Transformers, while outperforming standard patch-based inference and other block-convolution methods in memory and latency.

---

> ### Author Response · Authors · 2025-11-25
> **Response to Reviewer YTrW (2/2)**
>
> Below, we address your specific questions:
>
> **Q1** The weight $\alpha$ is empirically determined per network to balance local detail and global context. The reported experiments consider a single, fixed alpha across all layers.
> * **Sensitivity Analysis:** We performed a sensitivity sweep (linked below) and found that performance is stable for $\alpha \in [0.7, 0.9]$, peaking typically at 0.8.
>     * Figure: [https://ibb.co/gbMxZpxw](https://ibb.co/gbMxZpxw)
> * **Layer Configuration:** The decision of which layers to slice is automated by Algorithm 1 (Appendix A.3) based on the hardware memory constraint ($M_{max}$).
> * **Impact of $G_A$:** In Table 4 (Appendix A.4), we show that for moderate reductions, using only $L_E$ ($\alpha=1.0$) is sufficient. However, for high compression (e.g., MobileNetV2 at 16 slices), adding $G_A$ (reducing $\alpha$) can help recover accuracy and global context.
>
> **Q2** We found that strict consistency is less critical than **complementarity**.
> * **Role Separation:** $L_E$ captures high-frequency spatial details, while $G_A$ captures low-frequency semantic context.
> * **Fusion:** We use a simple weighted mean. While simple, this method is sufficient to outperform existing baselines (Table 2) without incurring the high computational cost of complex fusion modules (like attention-based fusion) that would bottleneck an MCU. Agin, while we propose a general and simple implementation, more advanced fusion mechanism (e.g. transformer based) could be investigated, especially for applying GaLe at the network design phase instead of post-training only.
>
> **Q4** We can confidently confirm there is no risk of dynamic leakage or environmental drift because GaLe is a static, offline optimization. The slicing, calibration, and graph restructuring happen only once on a host computer. The output is a standard static graph (e.g., ONNX) where tensor sizes are fixed. No dynamic memory allocation occurs during inference on the device. Since the graph structure and weights are frozen before deployment, temperature or voltage fluctuations cannot alter the inference logic or accuracy.

---

### Official Review · Reviewer_E4Fc · 2025-10-26

**Soundness:** 1
**Presentation:** 1
**Contribution:** 2
**Rating:** 2
**Confidence:** 4

**Summary:**

This paper introduces GaLe, a method designed to reduce the RAM consumption of deep neural network models for image processing during inference. The key idea is to approximate the outputs of trained network layers using two lightweight representations: a Local Exact (LE) map that preserves fine image details, and a Global Approximate (GA) map that retains global information. The paper demonstrates that GaLe can be applied to a wide range of architectures, including CNN-based models, transformer-based models, and hybrid CNN–transformer models. It also improves patch-based inference by adopting horizontal slicing for faster RAM access and further reduces computational overhead by choosing the minimal patch overlap needed to meet target accuracy. Experiment are done to show the efficiency and effectiveness of GaLe in inference for tasks including image classification, object detection, and image generation.

**Strengths:**

1. This paper introduces a new method to reduce the RAM consumption of image classification, object detection and image generation models during inference.

2. The proposed method is adaptable to a wide range of mainstream image processing models, such as CNN-based, and CNN-transformer based models.

**Weaknesses:**

1. The whole writing is very cryptic. Many details are missing.

	- There is no equation / formula / diagram to indicate how LE or GA is computed, making it impossible to understand how the method works.

	- The combination of LE and GA for hybrid models is not sufficiently motivated and not adequately justified. It remains unclear why the proposed LE and GA mappings can effectively approximate the outputs of the network layers.

	- There are many sentences like the following one in the text: "GaLe dynamically determines the number of patches for each block during the calibration pass, adapting to the memory footprint of the intermediate tensors" (Lines 245-246). However, there is no explanation on how it is done.

	- In Line 287, the paper claims that "our method can offer superior performance than approximation-based techniques." However, as far as I understand, the proposed method GaLe is also an approximation-based technique, as there is no formal proof to demonstrate that it achieves certain optimal performance.

	- In Line 175, it is argued that "accuracy is often more heavily impacted by other factors." However, performance degradation is resulted by the proposed method regardless of what other factors are. Explanations are needed.

	- In Line 323, the Id matrix and the unit matrix are essentially the same. Why do we need to use two different terms?


2. Concerns about the experimental evaluation.

	- How is the computational overhead reported in Table 2 and Figure 6 defined and measured?

	- Line 465 states that the proposed method achieves a 74% reduction for RT-DETR-L and an 88% reduction for YOLOv11n. How are these reduction percentages computed?

	- There is no sensitivity analysis of the important hyperparameter $\alpha$, which controls the proportion of the LE and GA mappings when approximating hybrid network outputs.


3. Some typos:

	- In Line 103, "the MobileNet Family..." instead of "te MobileNet Family...".
	- In Line 107, "and automated network..." instead of "an automated network..."

**Questions:**

Please kindly refer to the weaknesses.

---

> ### Author Response · Authors · 2025-11-26
> **Response to Reviewer E4Fc**
>
> Dear Reviewer,
> We sincerely appreciate the time you took to review our paper. We apologize if the presentation of the hybrid $L_E + G_A$ mechanism appeared cryptic.
>
> **W1:** We will revise Section 3 to explicitly formalize these definitions. GaLe is grounded in two established techniques:
> * **Local Exact ($L_E$) Formulation:** This component is an evolution of Partial Patch-Based Inference (PPBI). In PPBI, exact mathematical equivalence to standard inference is guaranteed if the overlap between patches equals the layer's receptive field ($O_{ppbi} = \mathcal{R}$).
>    * Our Modification: GaLe introduces an adaptive overlap $O_{GaLe}$. As described in Appendix A.3, we iteratively calculate the maximum possible overlap given the RAM constraint.
>    * Convergence: If the memory allows for $O_{GaLe} = O_{ppbi}$, our method converges to standard PPBI and becomes mathematically exact.
>    * Approximation: When memory is insufficient, $O_{GaLe} < O_{ppbi}$. Here, $L_E$ captures high-frequency details but loses global context at the boundaries.
> * **Global Approximate ($G_A$) Formulation:** To recover the global context lost by reducing the overlap, we use resolution scaling. $G_A$ is computed by downsampling the input, processing it, and upsampling the result.
> * **Combination:** The final approximation is a weighted sum, allowing us to trade off between the fine-grained detail of tiled inference and the semantic coherence of global downsampling.
>
> **Regarding Slicing:** We utilize horizontal (row-major) slicing. This is a hardware-aware design choice: on MCUs, this layout ensures memory contiguity for standard NHWC tensors, minimizing cache misses and DMA overhead compared to square tiling.
>
> **Optimal performance:** You are correct that GaLe acts as an approximation when memory is tight. However, the distinction we draw is that GaLe is asymptotically exact.
> * Pure approximation methods (like aggressive resolution scaling or token pruning) always degrade performance to save memory.
> * GaLe converges to the exact solution (PPBI) as memory permits.
>
> When we claim "superior performance," we refer to this flexibility: we achieve higher accuracy than pure downsampling (Table 2, "Res") and fit in tighter memory constraints than exact PPBI, allowing the user to choose a tradeoff between accuracy, overhead and RAM usage depending on target hardware.
>
> **Matrix Definitions** We apologize for any confusion caused by the terminology. In the paper we used:
> Id (Identity Matrix): A matrix with 1s on the diagonal and 0s elsewhere.
> All-ones matrix (J): A matrix where every element is 1.
> In Equation 3, these terms serve different purposes: $Id$ preserves the specific local values of the current block, while $J$ spreads the global average ($C_{ij}$) across the entire block. We will clarify the terminology in the revised version.
>
> **W2:** Computational overhead is defined as the ratio of additional Floating Point Operations (FLOPs) required by the slicing and merging mechanism compared to the original network's FLOPs.
> $$Overhead = \frac{FLOPs_{GaLe} - FLOPs_{Original}}{FLOPs_{Original}}$$
> As noted in the text, PPBI often incurs >100% overhead due to redundant re-calculation in overlap regions, whereas GaLe reduces this to <20% by optimizing the overlap.
> **Q:** How are these reduction percentages (74%, 88%) computed?
> These are derived from the hard constraints of the target hardware versus the model's requirements:
> * **RT-DETR-L:** Requires 9.8MB (INT8). Target hardware has 2.5MB.
> Reduction needed: $(9.8 - 2.5) / 9.8 \approx \mathbf{74.5}$%.
> * **YOLOv11n:** Requires 4.2MB (INT8). Target hardware has 512KB (0.5MB).
> Reduction needed: $(4.2 - 0.5) / 4.2 \approx \mathbf{88.1}$%.
>
>
> **Sensitivity Analysis on $\alpha$:** We agree this is critical. We have performed a sensitivity analysis on $\alpha$ for ImageNet classification.
> The plot at  [https://ibb.co/gbMxZpxw](https://ibb.co/gbMxZpxw) and in the revised appendix demonstrates that the method is stable for $\alpha$ values between 0.7 and 0.9.
> * $\alpha=1.0$: Corresponds to using only $L_E$ (pure slicing with insufficient overlap), leading to lower accuracy due to lost context.
> * $\alpha=0.5$: Relies too heavily on the low-resolution $G_A$, losing fine details.
> * Optimal Range: The peak at $\alpha=0.8$ confirms the necessity of the hybrid approach. We have added this analysis in the Appendix.
>
> **Performance degradation**. We agree that the degradation is real. Our argument in Line 175 is contextual: in embedded deployments (e.g., micro-robotics, edge sensors), the noise floor introduced by low-quality optics and analog sensors often degrades system-level accuracy by margins larger than the <1% drop introduced by GaLe. We do not mean to excuse the drop, but to contextualize its acceptability in the target domain

---

### Official Review · Reviewer_Tdm2 · 2025-10-31

**Soundness:** 3
**Presentation:** 3
**Contribution:** 2
**Rating:** 4
**Confidence:** 5

**Summary:**

The paper introduces GaLe, a novel, memory-efficient approximation technique for deploying large, pretrained deep neural networks on resource-constrained devices like microcontrollers without retraining. GaLe addresses the limitations of existing methods by using two complementary representations: a "Local Exact" ($L_E$) representation to preserve fine-grained details, and a "Global Approximate" ($G_A$) component to retain long-range dependencies. The "Local Exact" representation partitions the feature maps into multiple small tiles, which can significantly reduce RAM usage and computational overhead, while the "Global Approximate" helps maintain compatibility with modern architectures that use global operations and attention mechanisms. The authors demonstrate GaLe's effectiveness across various tasks, including image classification, object detection, and diffusion models, achieving performance comparable to exact inference but with substantial reductions in memory and latency, such as a 65% speedup on a Cortex-M33 core for a 90% RAM reduction compared to patch-based methods.

**Strengths:**

- This paper shows the real speedup and memory savings on different devices, demonstrating the effectiveness of the proposed methods.
- Evaluations across different tasks and models demonstrate the generalization of the GaLe method.
- GaLe can not only be applied to the convolutional neural networks, but also to the attention-based models without retraining.

**Weaknesses:**

- The novelty of the partitioning methods is limited. Such methods have been explored in previous architecture design works, e.g. [1] and [2]
- The technical details aren't clear enough. See the questions for more details.
- The paper could be further strengthened by including an analysis of how the results are influenced by different parameter settings. This would provide valuable insights into the sensitivity of the proposed method.

[1] Gang Li, et al, Block Convolution: Toward Memory-Efficient Inference of Large-Scale CNNs on FPGA, IEEE Transactions on Computer-Aided Design of Integrated Circuits and Systems, 2022

[2] Manoj Alwani, et al, Fused-Layer CNN Accelerators, MICRO, 2016

**Questions:**

- Does the attention-based model also need calibration? This paper only discusses the calibration for the convolutional layer to determine the overlap parameter. But for the attention layer, it seems that there is no overlap between different patches.
- There are many parameters during partitioning the feature map into different patches, e.g., patch size, overlap parameters, slicing patterns, and the weighting factor $\alpha$. For a given model, how to determine these parameters?
- This paper only evaluates GaLe on those vision tasks. Since GaLe can be applied to attention-based models, it would be better to evaluate the proposed method with LLMs.

---

> ### Author Response · Authors · 2025-11-26
> **Response to Reviewer Tdm2**
>
> Dear Reviewer,
>
> We sincerely thank you for your constructive feedback and for highlighting the practical value of our work. We have addressed your concerns regarding novelty, parameter sensitivity, and technical details below.
>
> **W1:** We thank the reviewer for pointing out these fundamental works on block-based processing. We acknowledge that tiled inference is a well-established concept for reducing memory in pure CNNs. However, GaLe was specifically developed to address the failure cases of these existing methods.
> * **Limitations of Prior Work:** Traditional block convolution methods (like those in [1]) fail when applied to modern architectures containing global operators. For instance, they cannot handle Squeeze-and-Excitation (SE) blocks or Self-Attention mechanisms because these layers require information from the entire feature map, which is broken during standard tiling
> * **GaLe's Contribution:** Our novelty lies in the Hybrid Decomposition ($L_E + G_A$). By introducing the Global Approximate ($G_A$) component, GaLe restores the global context lost during tiling. This allows us to extend memory-efficient inference to architectures where it was previously impossible, such as MobileNetV3 (SE blocks) and Hybrid Vision Transformers, while outperforming standard patch-based inference and other block-convolution methods in memory and latency.
>
> **Q1:** Your intuition is entirely correct. Unlike convolutional layers, our attention formulation processes distinct non-overlapping blocks of the Query, Key, and Value matrices. Therefore, no spatial overlap parameter is needed. For attention layers, the calibration step is used to determine the number of splits (blocking factor $b$) required to fit the calculation within the peak memory constraints, with similar effects as both the convolutional slice account and overlap parameters jointly. We will make this clearer in the revised version.
>
> **Q2, W3:** We determine parameters through a mix of hardware-constraints and empirical calibration:
> * **Hardware-Constrained Parameters (Slice Count & Overlap):** These are mathematically derived based on the device's RAM limit. As detailed in the pseudocode in Appendix A.3, the algorithm iteratively increases the number of slices until the memory footprint fits the target device
> * **Hardware-Optimized Slicing Pattern:** You raise an interesting point regarding the slicing strategy. We explicitly selected horizontal (row-major) slicing for the $L_E$ tiles to align with the memory layout of standard embedded runtimes (NHWC), ensuring data contiguity. This design choice minimizes cache misses and DMA transfer overhead, making it optimal for *latency* on resource-constrained devices. While we acknowledge that alternative patterns could prioritize accuracy over speed, similar to how we leveraged Token Merging (ToMe) instead of downsampling for the $G_A$ map in our Diffusion case study, these tend to be highly task-specific. We therefore prioritized a general, speed-optimal approach that serves as a robust baseline for developers
> * **Hyperparameters ($\alpha$ and Calibration Samples):** We have performed additional experiments regarding the weighting factor $\alpha$ and the calibration set size.
> **Value $\alpha$:** As shown in [https://ibb.co/gbMxZpxw](https://ibb.co/gbMxZpxw) and in the revised paper, the method is robust across a range of $\alpha$ values, and performance peaks around $\alpha=0.8$. We show how relying purely on local features ($\alpha=1.0$) or heavily on global approximations ($\alpha < 0.6$) yields suboptimal results. The combination ($0.7 \leq \alpha \leq 0.9$) provides the best trade-off, confirming the necessity of our hybrid approach.
> **Calibration Samples:** As shown in [https://ibb.co/N6V6211m](https://ibb.co/N6V6211m) and in the revised version, the calibration process is highly data-efficient. The Top-1 Accuracy stabilizes rapidly, requiring as few as 32 samples to reach optimal performance.
>
> **Q3** We appreciate the suggestion. However, we respectfully submit that LLMs are outside the scope of this specific work: GaLe is derived explicitly for Computer Vision tasks (Classification, Detection, Generation), exploiting the specific spatial locality inherent in image feature maps. In particular, our proposed way of obtaining the $G_A$ feature map heavily relies on redundancy in contiguous areas of the image/ feature maps, which may not be true for language tokens. We believe the current evaluation spanning CNNs, Hybrid ViTs, Object Detection (COCO), and Diffusion Models (Image Generation) provides a comprehensive validation of the method's core claims within the visual domain. Nevertheless, we see promising opportunities for future work exploring whether GaLe’s principles can inspire analogous mechanisms in sequential or multimodal architectures like LLMs and VLMs.

---

### Official Review · Reviewer_Vkmd · 2025-11-01

**Soundness:** 3
**Presentation:** 2
**Contribution:** 2
**Rating:** 4
**Confidence:** 3

**Summary:**

This paper proposes a memory-efficient inference framework GaLe for deep neural networks, aimed at embedded and resource-constrained devices. GaLe decomposes feature maps into two complementary representations: Local Exact (LE) that preserves fine-grained details via full-resolution features, and Global Approximate (GA) that retains long-range dependencies via low-resolution features. GaLe maintains compatibility with global receptive field operations and attention mechanisms, including hybrid CNN–transformer architectures without retraining, requiring only a lightweight calibration phase.

**Strengths:**

1. This paper is well-structured and presents the technical contributions in a clear and accessible manner. The proposed memory-aware local exact feature map slicing and global approximation techniques are straightforward and practical to implement.

2. The algorithms are technically solid and can be naturally extended to attention mechanisms, enabling their integration into transformer-based architectures and offering comprehensive insights into memory-efficient inference. It possesses a unified theory and practical framework across CNNs, transformers, and hybrid models.

3. Experimental results on ImageNet demonstrate that the proposed method significantly reduces RAM usage and improves inference efficiency compared to baseline approaches.

**Weaknesses:**

1. Although the proposed method is technically sound, it is not fully convincing that the inference slicing strategy for local exact feature maps is optimal. The approximation–accuracy trade-off is empirically tuned through calibration, and the work would be strengthened by a more rigorous theoretical analysis or formal characterization of the associated error bounds.

2. The comparison with prior work primarily focuses on methods such as PPBI and FPBI, which were proposed several years ago. Including more recent state-of-the-art approaches of training-free memory efficient methods, such as post-training quantization/pruning baselines in the evaluation would provide a stronger and more up-to-date demonstration of the effectiveness of the proposed method.

3. The experimental evaluation is limited to ImageNet dataset. To better demonstrate the generalization capability of the proposed method, it would be beneficial to include results on additional datasets or domains.

**Questions:**

1. Could you provide some theoretical analysis that why the inference slicing with learned padding is optimal for memory efficiency?

2. Could you provide some comparison with post-training quantization / pruning state-of-the-art methods?

3. Could you provide more comparison results on the other datasets, such as Places, iNaturalist, COCO, etc.?

---

> ### Author Response · Authors · 2025-11-26
> **Response to Reviewer Vkmd**
>
> We genuinely appreciate the insightful feedback you provided. Below, we address your specific concerns and questions.
>
> **Q1:** Thank you for your question. As detailed in Section 3.1 and the pseudocode in Appendix A.3, GaLe is designed to be adaptive, maximizing the slice size (and consequently the overlap) based on the available RAM. We will revise the final paper to make this analysis clearer.
> * **Convergence to Exact Inference:** In scenarios where the memory constraint allows for an overlap equal to the layer's full receptive field ( $O_{GaLe} =O_{ppbi}$ ), our method mathematically converges to standard Partial Patch-Based Inference (PPBI). In this state, the approximation becomes an exact computation with zero performance drop.
> * **Optimality under Constraints:** When the memory limit prevents full receptive field overlap, GaLe calculates the maximum possible slice size that fits within the specific hardware constraint ($M_{max}$). This is optimal in the sense that it maximizes the contribution of the Local Exact ($L_E$) features, which preserve fine-grained details, up to the exact limit of the hardware. Adding the $G_A$ feature map can additionally mitigate the errors in the cases where high memory compression is needed (see Table 4 in the supplementary materials)
> * **Global Receptive Field operations:** reducing memory of global receptive field operations requires heavy computation overhead due to the need to recompute previous results for each new data point (e.g., for memory-efficient attention algorithms). GaLe allows the user to select the desired tradeoff between accuracy and overhead, converging to standard memory-efficient attention when the allowed error is set to 0
> We will revise Section 3 to make this limit behavior, where GaLe can converge to PPBI or memory-efficient attention, more explicit.
>
> **Q2:** Thank you for raising this important point. GaLe is designed to be orthogonal and complementary to quantization and pruning, rather than a competitor to them. GaLe acts as a deployment engine that can further reduce the peak memory of models that have already been quantized or pruned. In fact, our experimental results explicitly utilize quantized models to demonstrate this compatibility - eg, Figure 5 and Section 6.1 explicitly mention quantized models. GaLe enables quantized models to run on devices where INT8 quantization alone is not enough to fit the peak memory requirements. We will update the experimental sections to clearly highlight this.
>
>
> **Q3:** We completely agree that demonstrating generalization is crucial. While the main results focus on ImageNet, we evaluate GaLe on other domains:
> * **Object Detection (COCO):** In Section 6.1 (Case Study: Object Detection) and Figure 6, we present results on COCO for RT-DETR-L and YOLOv11n. GaLe achieves negligible mAP reduction on these detection tasks while maintaining minimal computational overhead, vastly outperforming standard patch-based inference. Below we report additional results for YOLO11n at different configurations:
> | Split | Small AP | Medium AP | Large AP | RAM Usage (KB) |
> | :--- | :--- | :--- | :--- | :--- |
> | 4 | 19.8 | 42.8 | 56.8 | 1207.50 |
> | 8 | 19.6 | 42.4 | 56.6 | 603.75 |
> | 16 | 18.8 | 41.2 | 54.9 | 301.88 |
> | 32 | 17.5 | 39.5 | 53.1 | 150.94 |
>
> * **Fine-grained Classification (iNaturalist):** We report here additional results obtained by applying GaLe to a pretrained ViT on iNaturalist to evaluate fine-grained classification performance. We can see that the approach can keep high performance even in this task, with high RAM reductions and minimal performance loss.
> | Split | Accuracy | RAM Usage (M) |
> | :--- | :--- | :--- |
> | 1 | 91.98 | 6170.89 |
> | 2 | 91.797 | 3396.60 |
> | 4 | 90.732 | 1851.10 |
> | 16 | 89.859 | 519.27 |
>
> * **Image Generation:** To further demonstrate versatility beyond classification, we included a case study on Diffusion Models in Appendix A.1. Here, we apply GaLe to SD-Turbo, evaluating it using FID and LPIPS scores. As shown in Figures 7 and 8, GaLe allows for extreme memory compression while preserving high image fidelity. We report here some additional results for high-resolution image generation (RAM usage vs LPIPS score compared to the uncompressed model)
> | Attn split | LPIPS (512x512) | RAM (GB) | LPIPS (768x768) | RAM (GB) | LPIPS (1024x1024) | RAM (GB) |
> | :--- | :--- | :--- | :--- | :--- | :--- | :--- |
> | **1** | 0.00 | 5.77 | 0.00 | 12.98 | 0.00 | 23.07 |
> | **4** | 0.15 | 1.44 | 0.14 | 3.24 | 0.13 | 5.77 |
> | **9** | 0.16 | 0.64 | 0.20 | 1.44 | 0.09 | 2.56 |
> | **25** | 0.30 | 0.23 | 0.21 | 0.52 | 0.15 | 0.92 |
>
>
> We believe these results cover a wider breadth of generalization (Classification $\to$ Detection $\to$ Generation) than simply adding more classification datasets.

---

### Author Response · Authors · 2025-12-04
**Summary of Revisions and Additional Results for the Area Chair**

We sincerely thank the reviewers for their constructive feedback and the (new) AC for stepping in during this challenging time.

We provide here a short summary of the note strengths, main changes and additional experiments we added thanks to the reviewers’ feedbacks.

**Strengths Noted by the Reviewers**

 * Significant Practical Impact: Reviewers highlighted the "real speedup and memory savings" on embedded devices (Reviewer Tdm2), calling the 90% RAM reduction and 65% speedup "highly convincing" (Reviewer YTrW).
 * Broad Applicability: The method was praised for being a "unified framework" applicable to CNNs, Transformers, and Hybrid models (Reviewer Vkmd) without the need for retraining.
 * Versatility: Reviewers noted the method's effectiveness across diverse tasks, including classification, object detection, and diffusion models (Reviewer E4Fc, Tdm2).
 * Soundness: The approach was described as "technically solid" (Reviewer Vkmd) and "theoretically sound and practically applicable" (Reviewer YTrW).

**Rebuttal and Revision Summary:**
We have addressed the reviewers' questions and updated the manuscript with new data. The key revisions include:

 * Parameter Sensitivity & Calibration: We added a comprehensive analysis of the weighting factor $\alpha$ and the calibration overhead. The new results demonstrate that performance is stable across the recommended range ($\alpha \in [0.7, 0.9]$) and that calibration is highly data-efficient, stabilizing at ~32 samples (negligible offline overhead).
 * Generalization (Classification, Detection, Generation): To address requests for broader evaluation, we added a new classification dataset (iNaturalist). We also provided additional results for Object Detection and Image Generation, further validating GaLe's versatility across multiple domains.
 * Relationship to Token Merging (ToMe): addressing Reviewer YTrW, we clarified the compatibility of GaLe with ToMe. In our Diffusion Model case study, we demonstrate that replacing standard scaling with ToMe for the Global Approximate component allows for extreme memory compression while maintaining high image fidelity even at high resolutions.
 * Formalization: To address feedback on presentation, we clarified that GaLe is asymptotically exact when the memory budget allows for overlap $O \to$ Receptive Field, with GaLe mathematically converging to standard Partial Patch-Based Inference (PPBI) with zero error

We sincerely thank all the reviewers for the time and effort spent evaluating our work, for the thoughtful responses provided, and for their work for the ICLR community :)

---

### Meta-Review · Area_Chair_17sq · 2026-01-07

**Summary:**

This paper proposes  GaLe, a memory-efficient inference framework, for deep neural networks deployed on embedded and resource-constrained devices. The four reviewers pointed out several critical concerns regarding different aspects of this paper, including but not limited to:
1. The novelty of the proposed method is limited. Similar methods have been explored in previous architecture design works, such as Block Convolution.
2, The inference slicing strategy for local exact feature maps is not optimal. The approximation–accuracy trade-off is empirically tuned through calibration, and the work would be strengthened by a more rigorous theoretical analysis or formal characterization of the associated error bounds.
3. The compared baselines are too old. Including more recent state-of-the-art approaches of training-free memory efficient methods, such as post-training quantization/pruning baselines in the evaluation, would provide a stronger and more up-to-date demonstration of the effectiveness of the proposed method.
4. The experimental evaluation is limited to ImageNet dataset. It would be beneficial to include results on additional datasets or domains.
5. The technical details are not clear enough. There is no sensitivity analysis of the hyperparameters, such as $\alpha$.
6. The analysis of how the results are influenced by different parameter settings is missing. Such an analysis would provide valuable insights into the sensitivity of the proposed method.
7. The whole presentation is very cryptic. Many details are missing, such as no formula or diagram to compute LE or GA.
8. Under high memory compression ratios, the task-specificity of accuracy loss is not deeply analyzed. For example, in scenarios sensitive to details such as small object detection and high-resolution diffusion generation, whether the accuracy drop remains controllable.
9. The calibration phase adjusts the slice overlap (O) and slice number (N) iteratively to control errors, but the relationship between "calibration time" and "dataset size" is not explained. Given the limited computing power of embedded devices, excessive calibration time may undermine their engineering practicality.

The rebuttal addressed some of the concerns, such as the experiments on more datasets and the influence of hyperparameters. But the others remain unsolved. Further, this paper needs a thorough revision and cannot be accpeted in the current form.

**Reviewer Concerns:**

The 4th, 6th concerns are partially addressed, while the others are still outstanding.

**Reviewer Scores:**

None.

---

### Decision · Program_Chairs · 2026-01-26

Reject